# Terahertz response of monolayer and few-layer WTe$_2$ at the nanoscale

Ran Jing [1 ✉], Yinming Shao [1], Zaiyao Fei [2], Chiu Fan Bowen Lo[1], Rocco A. Vitalone[1], Francesco L. Ruta [1,3], John Staunton[1], William J.-C Zheng[1], Alexander S. Mcleod [1], Zhiyuan Sun [1], Bor-yuan Jiang[4], Xinzhong Chen [5], Michael M. Fogler[4], Andrew J. Millis[1,6], Mengkun Liu [5,7], David H. Cobden [2], Xiaodong Xu[2,8] & D. N. Basov [1]

Tungsten ditelluride (WTe$_2$) is an atomically layered transition metal dichalcogenide whose physical properties change systematically from monolayer to bilayer and few-layer versions. In this report, we use apertureless scattering-type near-field optical microscopy operating at Terahertz (THz) frequencies and cryogenic temperatures to study the distinct THz range electromagnetic responses of mono-, bi- and trilayer WTe$_2$ in the same multi-terraced micro-crystal. THz nano-images of monolayer terraces uncovered weakly insulating behavior that is consistent with transport measurements. The near-field signal on bilayer regions shows moderate metallicity with negligible temperature dependence. Subdiffractional THz imaging data together with theoretical calculations involving thermally activated carriers favor the semimetal scenario with $\Delta \approx -10\,\mathrm{meV}$ over the semiconductor scenario for bilayer WTe$_2$. Also, we observed clear metallic behavior of the near-field signal on trilayer regions. Our data are consistent with the existence of surface plasmon polaritons in the THz range confined to trilayer terraces in our specimens. Finally, data for microcrystals up to 12 layers thick reveal how the response of a few-layer WTe$_2$ asymptotically approaches the bulk limit.

[1] Department of Physics, Columbia University, New York, NY, USA. [2] Department of Physics, University of Washington, Seattle, WA, USA. [3] Department of Applied Physics and Applied Mathematics, Columbia University, New York, NY, USA. [4] Department of Physics, University of California, San Diego, La Jolla, CA, USA. [5] Department of Physics and Astronomy, Stony Brook University, Stony Brook, NY, USA. [6] Center for Computational Quantum Physics, Flatiron Institute, New York, NY, USA. [7] National Synchrotron Light Source II, Brookhaven National Laboratory, Upton, NY, USA. [8] Department of Material Science and Engineering, University of Washington, Seattle, WA, USA. ✉email: rj2466@columbia.edu

The physical properties of the enigmatic material tungsten ditelluride (WTe₂) depend critically on the number of layers. Bulk WTe₂ is postulated to be a type-II Weyl semimetal[1,2] with Fermi-arc surface states. Monolayer WTe₂ has been predicted and experimentally confirmed to be a quantum spin hall insulator[3–8] and exhibits gate-induced superconductivity[9,10]. Bilayer WTe₂ has broken inversion symmetry and is known to be ferroelectric[11], yet experiments produce ambiguous results on whether its electronic structure is semimetallic or semiconducting. Transport measurements support the semiconductor picture with a narrow electronic bandgap (<10 meV)[3]. Angle resolved photo-emission spectroscopy (ARPES), however, revealed that bilayers could also be weakly semimetallic with a small negative gap[12]. A combination of inverted bands, strong spin–orbit coupling and low crystal symmetry also makes few-layer WTe₂ an ideal system for studying topological effects such as the nonlinear anomalous Hall effect[13–15] and various unusual photogalvanic effects[16–18]. The goal of the present study is to explore the evolution of the low-energy electrodynamics of WTe₂ from monolayer to few-layer variants (Fig. 1a). We conclude that trilayer and thicker specimens are metallic and host surface plasmon polaritons (SPP)[19,20] that dominate the response in the terahertz (THz) range. The metallic response is reduced in bilayer areas and disappears in monolayer regions.

Bulk WTe₂ exhibits high electronic mobility and its intraband (Drude) optical response is entirely contained in the THz region[21,22]. Despite tremendous interest, the THz response of monolayer and few-layer samples remains unexplored. THz experiments on few-layer WTe₂ specimens are challenging because of the minuscule size of available samples typically under $10 \times 10\ \mu m^2$. The wavelength of THz waves is of the order of ~300 μm and conventional diffraction-limited methods are inadequate for interrogating the THz response of WTe₂ micro-crystals. In order to overcome the diffraction limit in THz, we utilize a scattering-type THz scanning near-field optical microscope (THz-SNOM)[23–27]. This technique is a hybrid of an atomic force microscope (AFM) with a pulsed THz source. AFM-based THz nanoscopy offers a robust experimental approach to investigate materials with sub-diffractional spatial resolution down to $\lambda/2000$ where $\lambda$ is the wavelength of the probe beam. THz-SNOMs are being successfully applied to an expanding list of materials and interesting problems. For example, THz-SNOM

methods have provided insights into nanoscale studies of electronic phase separation in the vicinity of the insulator-to-metal transition in VO₂[23,28,29], the plasmonic response of graphene[26–31], free carrier distributions in nanodevices[32,33], and phonon resonances in multiferroic materials[34].

Here we report on near-field nano-optical experiments in THz range for WTe₂ conducted at cryogenic temperature. The nano-THz measurements reveal that trilayers of WTe₂ show metallic behavior and a plasmonic response consistent with the properties of bulk crystal, whereas bilayer samples exhibit weak semimetallic behavior.

## Results

**THz near-field nano-imaging.** We investigated multi-terraced microcrystals of WTe₂ using a home-built apparatus enabling nano-THz experiments at cryogenic temperature[23]. The THz beam is focused onto an AFM tip with an 80 μm long shaft made of PtIr wire. The tip apex locally confines and enhances the THz electric field. The tip shaft functions as an antenna[35] and out-couples the near-field radiation into far-field radiation reaching the photo-conductive antenna (PCA) detector. The tapping of the tip modulates the near-field signal at ~70 kHz. We demodulated the amplitude of the tip-scattered electric field at the first ($S_1$) and the second ($S_2$) harmonics of the tip tapping frequency to suppress the undesired far-field background[23,36].

The exfoliated micro-crystals of WTe₂ are encapsulated between 6 nm of hexagonal boron nitride (hBN) on top and 20 nm hBN on the bottom (Fig.1a). The exfoliated structure is assembled on top of a SiO₂/Si wafer. This sample hosts terraces of mono-, bi-, and trilayer WTe₂ within a $25 \times 25\ \mu m^2$ area. These terraces are evident in both the optical inspection image (Fig. 1b) and in the nano-THz scan displaying the contrast in the scattering amplitude of the THz signal (Fig. 1c). The topographic contrast of AFM scans has only limited utility in visualizing the terraces because this contrast is suppressed by the top encapsulating layer (Supplementary Note 1). We obtained the network of dashed lines in Fig. 1b, c using a combination of optical contrast and nano-THz contrast. We remark that the top layer hBN is thin enough that the evanescent field from the sample is still detectable with the help of the AFM antenna tailored for the THz range.

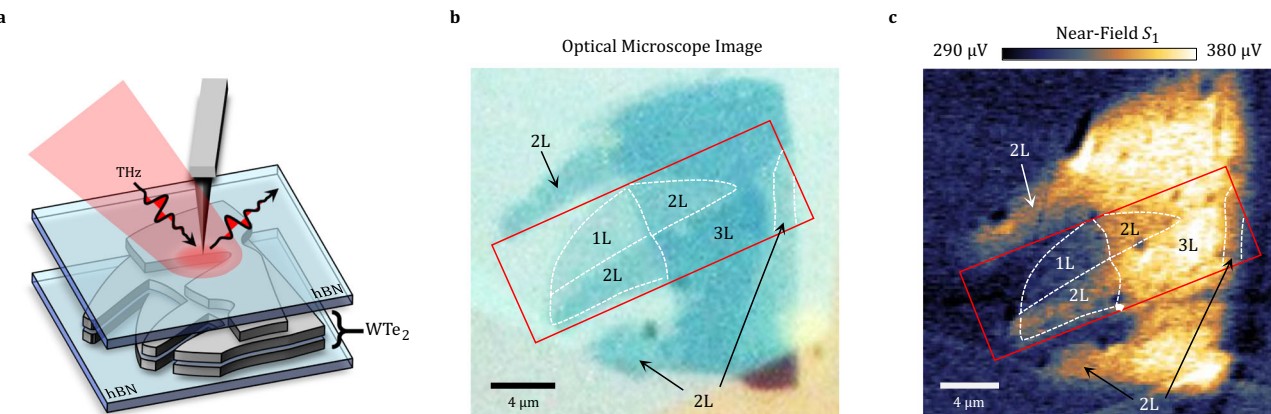

**Fig. 1 Schematic of nano-THz experiments on multi-terraced crystals of WTe₂. a** Metallic AFM tip locally enhances the electric field and enables THz coupling to materials at length scales much smaller than the THz wavelength. The size of the focused THz beam in the schematic is much smaller than the real focus. **b** Optical microscope image of the WTe₂ sample. Multi-terraced microcrystals of WTe₂ are encapsulated on top and bottom with hexagonal boron nitride (hBN) and reside on a SiO₂/Si substrate. Optical inspection reveals 1L, 2L, and 3L regions. The red frame indicates the field of view used for temperature dependent study in Fig. 2. We demarcate the boundaries of terraces labeled by layer number (1L, 2L, 3L) with white dashed lines. **c** THz near-field signal $S_1$ at room temperature, showing much higher THz signal in the 3L region compared to 1L and 2L.

In nano-THz experiments, the near-field scattering amplitude is an observable carrying information on spatially localized electro-magnetic response[36–39]. The measured signal is denoted by $S_{1,2} \propto \left| \widetilde{E}^{NF} \right|$ where $\widetilde{E}^{NF}$ is the THz near-field electric field. We analyzed the so-called approach curves: the variation of the $S_{1,2}$ signal as a function of separation between the tip and the sample (see Supplementary Note 2). This analysis confirmed that over 90% of $S_2$ originates from the near-field tip-sample interaction within 150 nm above the sample surface[40]. Demodulation of the THz signal at higher harmonics is not practical in view of the rapidly diminishing signal-to-noise ratio already at the third harmonic. The far-field contribution is enhanced at higher optical frequencies outside of the THz range[36]. For that reason, nano-optical experiments conducted in the mid-IR and visible ranges typically require demodulation at the third, fourth or even fifth harmonics[36]. In our nano-THz experiments, the tip radius is $R = 150 \sim 200$ nm as determined by scanning electron microscopy. The tapping amplitude is ~150 nm. The tip radius and the tapping amplitude govern the center momentum ($0.1/R \sim 1/R$) for photon scattering by the tip[41,42] and the achievable spatial resolution[40].

Here we report nano-THz imaging data collected in frequency-integrated mode at every pixel. The frequency range of the THz radiation in our experiments spans between 0.2 THz and 2.5 THz. Due to the antenna resonance effect of the tip, the near-field signal intensity is peaked at ~0.6 THz[23,43]. Our nano-THz apparatus is designed to produce hyperspectral images with frequency resolved information at every pixel by Fourier transforming the time-domain spectra[44]. However, frequency-integrated or "white-light" (WL) THz imaging has an important advantage of significantly increasing the signal-to-noise required to produce high fidelity images of weakly absorbing few-layer WTe$_2$ samples (see figures). We accompany nano-THz data with images in the infrared range where we employ a monochromatic light source (Supplementary Note 5).

In Fig. 2, we show the complete set of temperature dependent THz nano-imaging data. We plot the scattering amplitude signals $S_1$ and $S_2$ normalized by those of the SiO$_2$/Si substrate $S^{sub}$: $S_1/S_1^{sub}$ and $S_2/S_2^{sub}$. The $S_1$ data have a roughly two times higher signal-to-noise ratio (SNR) than $S_2$. Both $S_1$ and $S_2$ images display the same salient features. Since $S_2$ has less contribution from far-field background, we rely on $S_2$ to quantify the temperature dependence of the near-field response in the analysis that follows. We confirmed that the near-field signal due to the SiO$_2$/Si substrate shows negligible temperature dependence. We therefore can use the signal produced by the bare substrate as a reference in our normalization procedure. In all THz images, we resolve a feature due to a ~200 nm wide topographic linear defect marked in the panel obtained at 100 K. This latter topographic feature confirms that the spatial resolution of our THz near-field imaging is well below ~200 nm at all temperatures.

To analyze the contrast between terraces with different numbers of WTe$_2$ layers, horizontal line-cuts from the $S_1$ images are displayed in Fig. 2d. The location of the line-cut is indicated

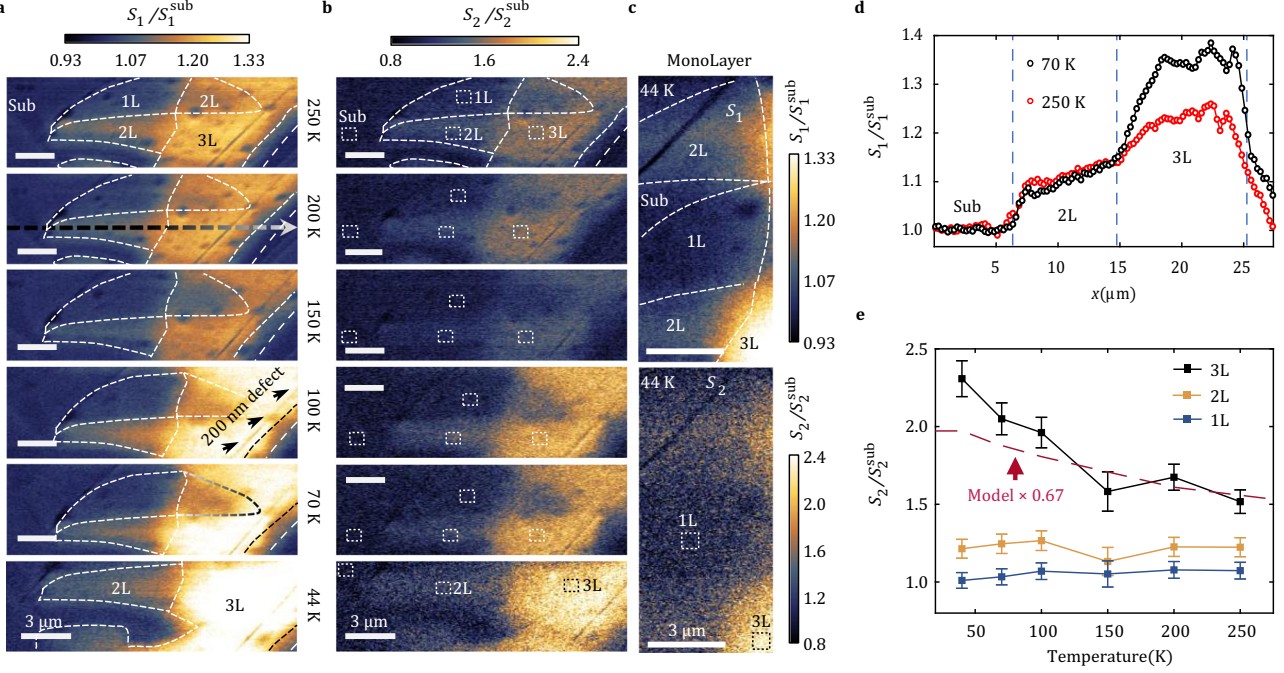

**Fig. 2 Temperature-dependent near-field maps of nano-THz response of WTe$_2$ micro-crystals.** The scale bars in all panels are 3 μm. **a** Near-field images of the normalized amplitude contrast $S_1/S_1^{sub}$ of nearly identical regions at 6 different temperatures between 250 K and 70 K. The 44 K image includes only 2L and 3L regions and the 1L region is shown in Fig. 2c. Broadband THz signal utilized in these images shows intensity peaked at 0.6 THz. The signal due to the SiO$_2$/Si substrate ($S_1^{sub}$) has negligible temperature dependence in the THz range studied here. The horizontal dashed arrow in the panel at 200 K indicates the scanning line-cut used to construct the plot in Fig. **d**. Micrometer-sized dark spots also visible in the topographic AFM contrast can be attributed to bubbles in the encapsulated structures. **b** Near-field $S_2/S_2^{sub}$ images taken simultaneously with $S_1/S_1^{sub}$. **c** Enlarged images zoomed at the interface between 1 L and 2 L, 1 L and the substrate. **d** $S_1/S_1^{sub}$ line-cut (averaged over 5 neighboring pixels) at 250 K and 70 K. The line-cut corresponds to the arrow in Fig. 2a. **e** Normalized $S_2$ signals averaged in the regions indicated in the $S_2$ images (white dashed boxes) for the substrate and for 1L, 2L, 3L regions of WTe$_2$. The filled squares are experimental data. The error-bars are the standard deviations of the extracted data. The dashed line is a model calculation using the dielectric properties of the bulk material. The magnitude of the signal for bulk dielectric properties is rescaled by a factor of 0.67 to be comparable with 3L WTe$_2$.

with a dashed arrow in the 200 K image of Fig. 2a. The line-cut shows evident plateaus corresponding to terraces with different numbers of layers. Regions with a higher number of layers exhibit higher near-field signal. The signal in the trilayer region increased substantially at lower temperature, typical of metallic responses. Monolayers (Fig. 2c and Supplementary Note 3) are marginally distinguishable from the substrate, demonstrating a clear insulating response. Interestingly, the bilayer region produced an intermediate amount of signal. While the overall near-field signal is 10–15% higher than the insulating monolayer, the absence of any temperature dependence restricts the magnitude and sign of the bandgap, as we will discuss later. In addition to $S_1$, the $S_2$ signal was analyzed in small areas at the center of three different regions. These areas are indicated as white dashed rectangles in the image in Fig. 2b. The temperature dependence of nano-THz contrast extracted from this analysis is plotted in Fig. 2e. The signal in the trilayer area increases by more than 40% between ambient and 44 K, whereas in bilayer and monolayer regions, the increase of signal at low temperature is absent.

It is instructive to compare the temperature dependence of the THz near-field contrast summarized in Fig. 2d with DC transport data[3]. The DC conductivity of trilayer WTe2 is metallic at all temperatures in agreement with the nano-THz trend we report in Fig. 2d. For bilayer WTe2, DC transport data indicates a semiconducting behavior with a narrow gap in the meV range[3]. Specifically, the DC conductivity drops significantly below 100 K[3]. In many conducting materials, the real part of the optical conductivity in the THz range matches the DC value. However, if a material has a THz-range gap, this will not be the case. Indeed, the temperature independent nano-THz response of the bilayer terraces contrasts with the drop of the DC conductivity in undoped bilayer WTe2 at low temperature. We note that hBN encapsulated WTe2 is normally found to be almost undoped and therefore extrinsic doping of this sample is unlikely[3].

Nano-THz imaging data presented in the form of two-dimensional maps in Fig. 2a, b or line-cuts in Fig. 2d reveal a significant spatial dependence of the scattering signal. This effect is manifested as a gradual change in both the $S_1$ and $S_2$ signal within a 2–3 μm vicinity of the boundaries of trilayer WTe2 and across bilayer regions. We remark that the width of these transitional regions is significantly larger than the spatial resolution of our near-field imaging apparatus (~200 nm), as well as the width of the physical boundary observed in Fig. 1. Comparing the line-cut curves acquired at different temperatures in Fig. 2e, the location and the width of the transitional region has no noticeable dependence on temperature. With the help of

real-space near-field modeling of SPPs on the confined structure presented in the latter part of the paper, we show that the gradual spatial variation of the signal arises from THz SPPs with long wavelength (6–20 μm on trilayer).

We now discuss our data in the context of recent observation of edge states in WTe2[4,6]. In Fig. 2c, we further zoom in on the monolayer region at the lowest temperature, 44 K. If edge states produced contrast in the THz range, we would see signal near the boundary between the monolayer WTe2 and hBN/SiO2/Si substrate and possibly also at the boundaries between monolayer and bilayer terraces. Indeed, such signals near the boundary are seen in the GHz regime[4]. However, we observe no significant signal at the boundaries of the monolayer. This is likely due to that the conductance of the topological edge state is too low to induce observable contrast in near-field imaging. In addition, the contrast of the edge state could be suppressed if the width of the state is narrower than the resolution.

**Modeling of nano-THz response.** To understand the THz near-field contrast of WTe2 microcrystals, we carried out modeling of the response associated with the trilayer region. We assumed that trilayer WTe2 has the same relative permittivity as bulk WTe2[21]. This simple assumption allows us to determine the origin of the temperature dependence of the THz signal in the trilayer region. We will discuss the transition from trilayer to bulk WTe2 in terms of THz near-field response later in the text. In our analysis we consider encapsulating hBN layers as well as the response of the SiO2/Si substrate within the framework of the lightning-rod model (LRM), a multilayer model of the near-field response described in McLeod et al.[45]. An implicit assumption of the model in McLeod et al.[45] is that all layers in multi-layered structures are either isotropic or uniaxial with an out-of-plane optical axis. On the contrary, WTe2 reveals notable in-plane anisotropy with distinct plasma frequencies between a and b axes within the WTe2 plane[21,22]. In the analysis that follows, we assumed that trilayer WTe2 can be reasonably described as a uniaxial material with its in-plane relative permittivity represented by that of b-axis of bulk WTe2. We also performed calculations with both a purely a-axis response and an effective dielectric function averaging between a-axis and b-axis data. All three methods produce qualitatively similar results (Supplementary Note 4).

We proceed with the quantitative analysis of the nano-THz response of WTe2 trilayers by calculating the p-polarized reflectivity $r_p(\omega, q)$ following the procedure described in McLeod et al.[45]. The imaginary part of $r_p(\omega, q)$ (Fig. 3a) reveals a branch

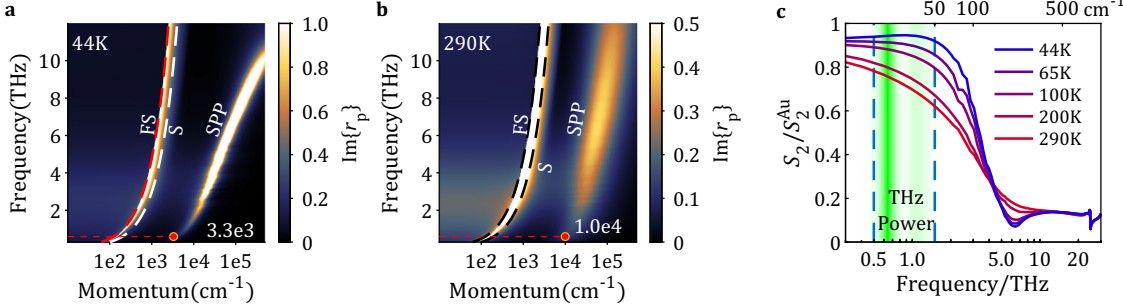

**Fig. 3 Electrodynamics and plasmonic response of 3L WTe2.** Modeled imaginary part of the momentum dependent p-polarized reflection coefficient $r_p(\omega, q)$ at **a** 44 K and **b** 290 K, based on bulk dielectric function data[21]. The red points located at $q = 3.3e3\,cm^{-1}$ ($\lambda = 19\mu m$) and $q = 1.0e4\,cm^{-1}$ ($\lambda = 6.3\mu m$) indicate the momentum of SPP at 0.6 THz. Free space (FS) and SiO2 (S) light lines are indicated with dashed lines, respectively. SPP dispersions are clearly observed, and the dispersion broadens with increasing temperature. **c** Spectra of the near-field scattering amplitude modeled following McLeod et al.[45] at different temperatures based on calculated $r_p(\omega, q)$. The shaded area indicates the frequency spectrum of our THz source.

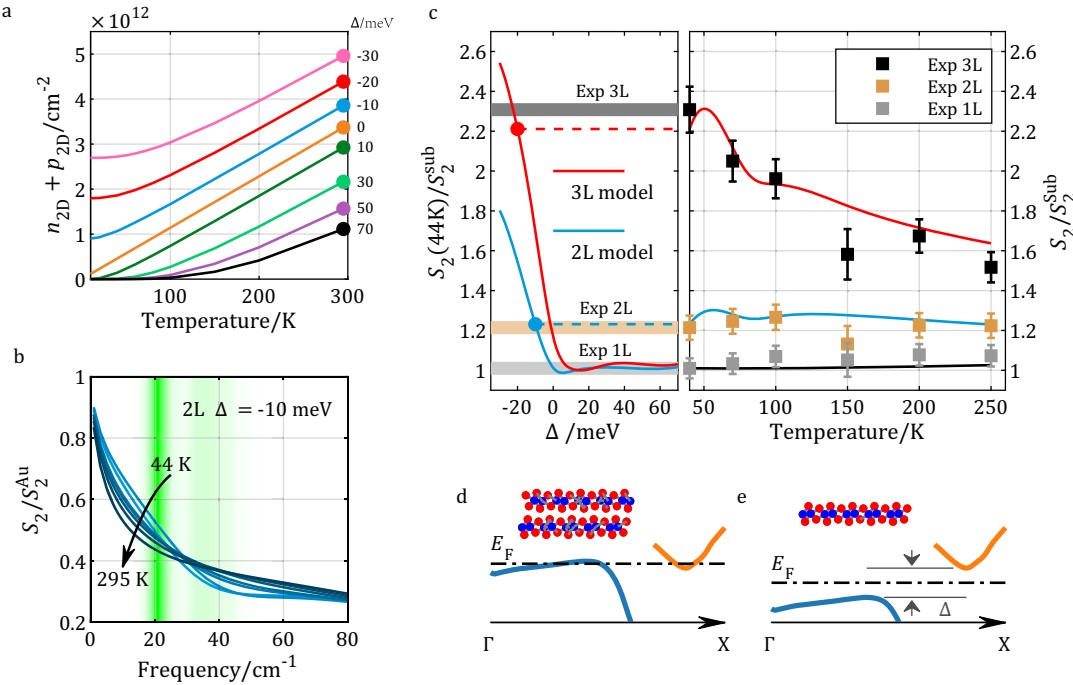

**Fig. 4 Near-field electrodynamics of thermally activated carrier of few-layer WTe₂. a** Temperature dependence of thermally activated carrier densities at different gap sizes calculated (Supplementary Note 6) based on the band structure investigated by ARPES [12]. **b** Near-field spectroscopic response of the thermally activated carriers of a model 2L WTe2 with $\Delta = -10$ meV. The green shaded region represents the power spectrum of the THz probe. **c** Right panel: temperature dependent WL signal calculation based on LRM for 1L (black curve), 2L (blue curve), and 3L (red curve) WTe₂ with gap sizes 60 meV, −10 meV, and −20 meV respectively. Along with the model, nano-THz data of 1L, 2L, and 3L WTe₂ are displayed with squares. The error-bars are the standard deviations of the extracted data in regions indicated in Fig. 2b. Both the model curves and experiment points are normalized to the substrate value. Left panel: the gap-size dependent near-field signal of 2L (blue) and 3L (red) at 44 K. The signal level is strongly suppressed when the gap is close to zero or positive. **d, e** Hypothetical band structure of semimetallic 2L WTe₂ (left) and insulating 1L WTe₂ [12] with a bandgap Δ>60 meV (right).

of strongly dispersing SPP. The three modes in Fig. 3a, b are, from left to right, the free-space light line, the light line in SiO₂ and the SPP in trilayer WTe₂. The SPP is sharp at low temperatures, while at 290 K it is overdamped. This is due to reduced scattering of electrons at low temperature[46,47].

The dispersion calculation in Fig. 3a, b implies that the SPP wavelength is 6–20 μm in the THz range. Because the tips we utilize in nano-THz experiments have radii $R = 150 \sim 200$ nm, we gain access to the range of momenta peaked around $\frac{0.1}{R} \sim 5 \times 10^3$ cm$^{-1}$[41,42,45] Since the THz intensity in our experiments is spread over 0.5–1.5 THz, we can extract the accessible range of wavelengths of the SPP modes from Fig. 3a. This straightforward procedure suggests that the relevant modes occur between $4 \times 10^3$ cm$^{-1}$ and $8 \times 10^3$ cm$^{-1}$, implying that the wavelengths of these modes span the range between 6 and 20 μm. Our THz near-field tip is thus expected to efficiently couple to SPP modes in trilayer WTe₂.

Next, we calculated the near-field spectra of WTe₂ based on $r_p(\omega, q)$ dispersion calculations. In Fig. 3b, we show the near-field amplitude spectrum produced within the framework of the LRM[45] at different temperatures. In the 0.5–1.5 THz range, the measured near-field signal is governed by the SPP of WTe₂. At low temperatures, plasmonic losses due to electron–phonon scattering are reduced and the SPP mode becomes more pronounced. By integrating the near-field signal at all frequencies investigated with our THz apparatus (shaded region in Fig. 3c), we acquired the model near-field signal at all temperatures. The result is plotted in Fig. 2e (red dashed line) along with the experimental data. This analysis captured the temperature

dependence of the experimental data but produces higher signal level than the measurement. Therefore, we conclude that the temperature dependence of 3L WTe₂ is impacted by the SPP. The fact that the model signal is overall higher indicates that 3L WTe₂ is less metallic than the bulk.

While the presence of a large gap of >60 meV in monolayer WTe₂ is demonstrated by transport[3] and ARPES[12] measurements, the semiconductor versus semimetallic nature of the bilayer remains unclear. ARPES experiments on bilayer WTe₂[12] indicate a vanishing, if not negative, gap (Fig. 4d). Transport measurements indicate semiconducting/insulating behavior with a small positive gap (<10 meV)[3] (Fig. 4e). Our local nano-THz experiments provide a unique probe in the relevant frequency region, without complications from electrical contacts and inevitable defects. The pronounced temperature dependence observed in metallic trilayers is partially due to the impact of the SPP. The complete insulating behavior of monolayer areas is likely due to its large gap (>60 meV). On bilayer WTe₂, the fact that its near-field signal is higher than monolayer WTe₂ requires a weak metallicity. Here we neglect interband optical absorption at THz frequencies due to the indirect gap of bilayer WTe₂.

Within the small gap or negative gap scenario, thermally activated carriers are the main contributor to the weak metallicity of bilayer WTe₂. In Fig. 4, we theoretically investigated the temperature dependence of the near-field signal due to the thermally activated carriers in bilayer WTe₂ with different gap sizes (Supplementary Note 7). In Fig. 4a, when the gap size is in the range of −10 meV to 10 meV, the carrier density at 44 K is as high as $n_{2D} = 0.2 \sim 1e12$ cm$^{-2}$, which is smaller than the value

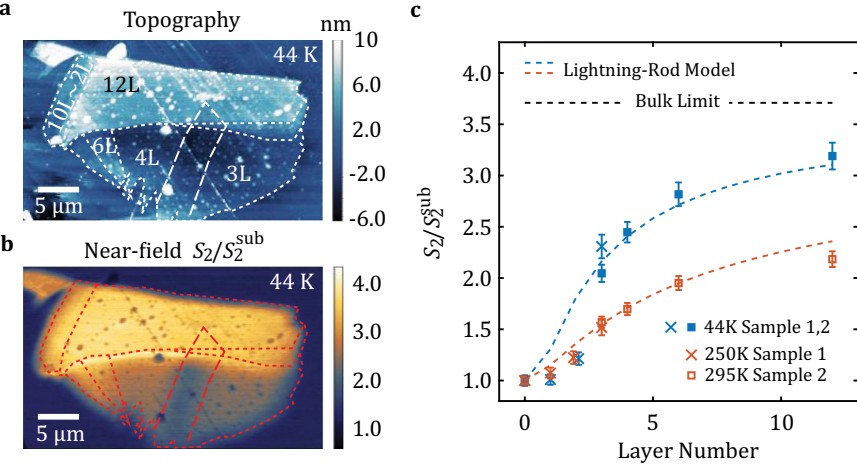

**Fig. 5 Layer-dependent WL near-field signal on WTe₂. a** Topography image of the sample at 44 K. The boundaries of different regions are highlighted with dashed lines. **b** Normalized WL signal $S_2/S_2^{sub}$ imaged simultaneously with the topography. **c** Layer-dependent near-field signal on few-layer WTe₂. The experiment data are collected on two different samples. The data points extracted from Fig. 2 (sample 1) are marked with crosses. The data points acquired on sample 2 are marked with squares. 3L WTe₂ is measured on both samples. High and low temperature data are displayed in red and blue points. The error-bars are the standard deviations of the extracted data in Figs. 2b and 5b. The dashed lines are LRM near-field signal contributed by thermally activated carriers with a gap size ~−20 meV at 44 K (blue) and 295 K (red).

($3.6e12\,\text{cm}^{-2}$) estimated in the ARPES experiment[12]. The temperature dependences of the scattering rate and of the carrier density dictate the temperature dependence of the near-field response. As is shown in Fig. 4b, thermally activated carriers directly contribute to the signal measured in our experiment.

At $\Delta \sim -10\,\text{meV}$, the simulated temperature dependence of WL near-field signal (blue curve in the right panel of Fig. 4c) matches the experimental data well. In Supplementary Note 7, the temperature dependence of the WL near-field signal corresponding to different gap sizes of bilayer WTe₂ are displayed. When the gap size is larger than 10 meV (~2.5 THz), thermally activated carrier density is sufficiently low that the near-field response in our THz range (0.5–1.5 THz) resembles an insulator. In the right panel of Fig. 4c, we modeled the temperature dependence for monolayer (black curve) with this "large gap" scenario ($\Delta = 60\,\text{meV}$). When the gap size is reduced below $+10\,\text{meV}$ (Supplementary Note 7 and Supplementary Fig. 6c), the near-field signal at high temperature gradually increases and is comparable to the experiment value. However, the carrier density (Fig. 4a) at low temperature gradually vanishes, leading to a strong suppression of the near-field signal at low temperatures (Supplementary Fig. 6c). The temperature independent behavior for bilayer WTe₂ observed in the experiment (orange square dots in Fig. 4c) therefore calls for a finite carrier density even at the lowest temperature (44 K), which favors the semimetallic scenario. Once the gap size is reduced to −10 meV (overlapping conduction and valence band), the signal at low temperature becomes comparable to that at high temperature and better fits the experimental value (Fig. 4c). In the left panel of Fig. 4c, we summarized the gap-size-dependent near-field signal at the base temperature 44 K. Further increasing the absolute negative gap leads to an increase of near-field signal at low temperatures to the levels exceeding data for bilayer WTe₂, due to the abundance of carriers (Supplementary Fig. 6c). Therefore, our observation of the temperature independent WL signal on bilayer WTe₂ favors the semimetallic nature with a small negative gap ($\Delta \sim -10\,\text{meV}$).

We applied the same calculation to the trilayer WTe₂. In the left panel of Fig. 4c, we summarized the gap-size-dependent near-field signal at the base temperature 44 K as well. Because of the

thickness effect, the simulated near-field signal on trilayer is higher than bilayer with the same gap size. In the right panel of Fig. 4c, the simulated temperature dependence of the WL signal on trilayer WTe₂ with $\Delta = -20\,\text{meV}$ (red curve) almost perfectly fits the experimental data. Therefore, thermally activated carriers with $-20\,\text{meV}$ gap better explain the trilayer near-field signal compared to the simulation using bulk WTe₂ optical constants (Fig. 3).

To illustrate how THz near-field signal evolves with thicker WTe₂, we prepared a different sample with 3L, 4L, 6L, and 12L WTe₂ [48] on which the same measurement was performed. Except for the difference in the thickness of WTe₂, the overall configuration of the sample is the same. In Fig. 5c, the data extracted from Fig. 2 (Exp 1) and the data extracted from Fig. 5b (Exp 2) are displayed side-by-side. With the increase of the layer number, the near-field signal increases rapidly from 3L and the growth rate decreases with the increase of the layer number. For 12L at 44 K, the near-field signal level is 86% of the bulk WTe₂ calculated using bulk optical constants[21]. According to Fig. 5, the growth continues at 12L, but the converging behavior is already obvious. By assuming 3L–12L WTe₂ can still be described by a 2D band structure, we apply the same model described in Fig. 4 on these thicknesses. Here, we fitted the 3L–12L experimental data at high and low temperature using LRM with only one free parameter, the gap size $\Delta$. When $\Delta \sim -20\,\text{meV}$, the model results simultaneously matched the high and low temperature data. Therefore, from 3 L to 12 L, WTe₂ can be reasonably described as a semimetal with a negative gap $\Delta \sim -20\,\text{meV}$.

**Modeling of polaritonic patterns in real-space**. With knowledge of the THz electrodynamic properties of mono-, bi- and trilayer regions on our WTe₂ microcrystal, it is now possible to model the real-space pattern (Fig. 6) of the THz near-field based on the geometry of the sample shown in Fig. 1c. Following the analysis in Fig. 4, we assigned a semimetal model with a −10 meV gap for bilayer and a −20 meV gap for trilayer regions. We adopted the permittivity extracted from DFT calculation[49] for the monolayer region. The real-space modeling in Fig. 6 considers the

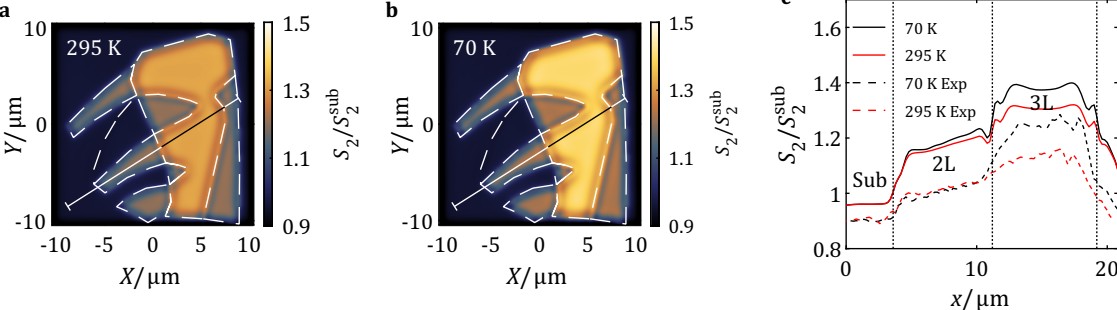

**Fig. 6 Real-space modeling of SPP pattern. a, b** Result of the model real-space near-field ($S_2/S_2^{sub}$) pattern associated with SPP at 295 K (**a**) and 70 K (**b**) on the investigated sample geometry. Dashed lines highlight the physical boundaries of all regions. Line-cuts across 2L and 3L WTe$_2$ are indicated by the solid line in both images. **c** Model $S_2/S_2^{sub}$ line-cuts extracted from **a** and **b** are plotted in solid lines. Along with the model result, experiment line-cuts are displayed in dashed lines and are shifted vertically for clarity.

intrinsic SPP mode on the experimentally measured geometrical configuration of the microcrystals. Further details of this real-space calculation are provided in the Supplementary Note 8.

The real-space near-field modeling results for few-layer WTe$_2$ (Fig. 6a, b) are in excellent agreement with the experimental images (Figs. 1c and 2a, b). In Fig. 6c, line-cuts were extracted at the same location with Fig. 2 and are compared with the experimental results. In the case of bilayer, the model shows the temperature independent behavior of the signal level as expected from the result in Fig. 4. The slope of the signal from the substrate side to the trilayer side is also reproduced well. Importantly, the gradual transition of near-field signal on bi- and trilayer edges are present in both experiment and model results, proving that the blurred edges are caused by the long wavelength of the THz range SPPs. In the model result, a weak fringe pattern can be recognized on 3L WTe$_2$. In real samples, however, the fringe signature could be easily erased due to the lower quality factor of SPPs. As for bilayer WTe$_2$, despite the low carrier density, the near-field response of SPPs can be detected in THz frequencies and is strongly impacted by the thermally activated carriers. According to Fig. 4c, a similar response is also expected in narrow gap semiconductors at even higher temperatures.

In conclusion, we investigated the low temperature nanoscale electromagnetic response of few-layer WTe$_2$ micro-crystals at THz frequencies. The low-temperature near-field signal has a strong dependence on the number of layers. The response of trilayer WTe$_2$ is clearly metallic as evidenced by the temperature dependence and is dominated by SPPs in the confined geometry of narrow terraces. The weak response of monolayer is consistent with an insulator with relatively large bandgap. Bilayer WTe$_2$ shows higher THz signal than insulating monolayers but the observed THz response is also independent of temperature from 250 K to 44 K. This latter behavior implies finite carrier density in bilayers down to the lowest temperature of this experiment (44 K). Comparison to our model suggests that the WTe$_2$ bilayer is a semimetal with a small negative gap $\Delta \sim -10$ meV for bilayer WTe$_2$. When the layer number is higher than three, the near-field signal continues growing and a negative gap $\Delta \sim -20$ meV can reasonably describe 3–12L WTe$_2$. For 12 L at 44 K, the near-field signal level is ~86% of the bulk WTe$_2$, calculated based on bulk optical constants. Finally, knowledge of the electrodynamics of mono-, bi-, and trilayer WTe$_2$ in our sample allows for a direct real-space modeling of the THz near-field signal, which matches the experiment well. Our complete temperature dependent THz near-field images together with theoretical modeling paves the

way for understanding the low energy electrodynamics of future quantum materials beyond the diffraction limit.

## Methods

**THz scanning-type near-field optical microscope**. Both the AFM scanner and focusing optics of our apparatus (Fig. 1a) are situated in an ultra-high vacuum (UHV) compartment[23]. This allows for measurements at temperatures down to ~40K limited by the imperfect thermal contact of a sample carrier introduced through rapid access load locks into our UHV system.

In the experiment on sample 1 (Figs. 1 and 2), we utilize a pair of low temperature-grown GaAs photoconductive antennas (PCA, Neaspec GmbH) as emitter and detector. We activate both PCAs with a 1550-nm femtosecond fiber laser after doubling its frequency in a nonlinear crystal. In the experiment on sample 2 (Fig. 5), we utilize optical rectification of a single pump beam for THz generation and electro-optic (EO) sampling for THz detection. By tilting the phase front of a 17 W, 1030 nm pump beam, we achieve the necessary phase matching condition to generate THz radiation via optical rectification in LiNbO3 with an efficiency of 0.1%. The scattered beam is routed to a ZnTe crystal for EO detection in the time domain using a delta-function like 800 nm gate beam with pulse duration of 20 fs.

In this experiment, we exploit the frequency-integrated (WL) signal to produce high fidelity images. When a THz pulse is scattered by the tip and reaches to the detector, we can measure this pulse at different time point $t_m$. If we tune $t_m$ to the main peak of the detected pulse where the phases of all frequency components in the wave packet are roughly equal, the WL signal is acquired. For trilayer WTe$_2$, the near-field spectra are almost flat (Fig. 3c). Therefore, WL images are suitable to track its temperature dependence. For bilayer and monolayer regions, because of the low signal level, WL images are needed to produce meaningful results.

**Preparation of WTe$_2$ microcrystal**. WTe$_2$ crystals are mechanically exfoliated onto highly p-doped silicon substrates consisting of 285 nm SiO$_2$[48,50]. WTe$_2$ flakes of mono- to trilayers are optically identified and encapsulated within hBN flakes using standard polymer-based dry transfer technique. The top and bottom hBN flakes used for encapsulation are typically 5–7 nm thick and 12–30 nm thick, respectively. Both WTe$_2$ exfoliation and encapsulation processes are performed inside a nitrogen glovebox (oxygen and water vapor levels are <0.5 ppm). The polymer on top of the heterostructures are dissolved outside the glovebox before near-field optical measurements.

**Lightning-rod model calculations of near-field signals**. We mainly follow the modeling procedure described in McLeod et al.[45]. The modeling is based on reflection coefficient $r_p(\omega, q)$ of the layered structure of the sample. A numerical solution to the electric field distribution of a tip-sample system is used to calculate near-field signal. In this way, parameters like tip radius and tapping amplitude is considered in the modeling. However, because the model is based on a 19-μm-long metallic tip with a cone structure. It does not account for the resonance of the 80-μm tip to THz beam in the experiment. Our solution is to manually multiply the model spectra with the spectra measured on Au and use it as an approximation to experiment result.

## Data availability
The data that support the findings of this study are available from the corresponding author upon reasonable request.

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

## Acknowledgements
Research on "Terahertz response of monolayer and few-layer WTe₂ at the nanoscale" is supported as part of Programmable Quantum Materials, an Energy Frontier Research Center funded by the U.S. Department of Energy (DOE), Office of Science, Basic Energy Sciences (BES), under award DE-SC0019443. The Flatiron Institute is a division of the Simons Foundation.

## Author contributions
D.N.B., R.J., and Y.S. conceived the experiments. R.J. and Y.S. performed the THz near-field imaging experiments. R.J. and R.A.V constructed the THz near-field device and beam line. Z.F., D.H.C., and X.X. fabricated the WTe₂ devices. R.J., F.L.R., J.S., A.S.M., B.J., and M.M.F. conducted the lightning-rod modeling. C.F.B.L, W.J.-C.Z., and A.S.M. performed real-space near-field modeling. Z.S., X.C., A.J.M. and M.L. provided helpful comments on the interpretation of the data. R.J., Y.S., and D.N.B. wrote the paper with input from all coauthors. D.N.B. supervised the project.

## Competing interests
The authors declare no competing interests.
