## [Peer Review File · Nature Communications]

REVIEWER COMMENTS

Reviewer #1 (Remarks to the Author):

The authors apply a (fairly) novel technique to exfoliated WTe₂ flakes in the 1-3 monolayer regime. Much of the work addresses validation of the method in the context of this material system.

The data the authors receive are impressive, especially given the novelty of the technique. The data requires, however, modeling to gauge the importance of lateral surface plasmon polaron scattering, which render the 'steps' the authors describe far more gradual than maybe desired. The problem here is NOT one in which the data obtained by the authors is questionable, but rather one, in which edge effects – and indeed many localized effects this technique would otherwise be a great tool to study – are obscured.

What is left is a characterization of the material across an interesting thickness range that is complementary to existing transport and spectroscopic measurements. Enhanced utility of their technique derives maybe from applicability to smaller samples. There is little or no concern that the data obtained by the authors is valid.

Overall, the manuscript offers information on the films that is well in line with common expectations. There is very little new or outstanding about the paper, other than the application/development/validation of a new technique. This paper should definitely be published. It ought to be an editorial decision, whether Nature Communication is the proper venue for data as described above.

Reviewer #2 (Remarks to the Author):

Authors report THz range EM behaviors of WTe₂ mono-, bi-, and tri-layers, encapsulated in hBN, as a function of temperature. Authors observe / conclude a metallic state and SPPs for the tri-layer region, moderate metallicity for the bi-layer region, and weakly insulating behavior for the monolayer region. The main data is presented in Figure 2 and the manuscript relies heavily on modeling to understand the data. I have several technical comments that I would like the authors to address.

#1. Seeming discrepancy between Fig. 2d and the S2 image (Fig. 2b).

The S2 image at 44 K looks shifted vertically. It looks shifted because the feature corresponding to the 2L region becomes suddenly dark at 44 K and the 1L region becomes brighter at 44 K. Thus, comparing the 1L region of 150 K versus 44 K for example, the normalized S2 averaged signal of 44 K should be higher than that of 150 K based on the images. And for 2L region, the normalized S2 averaged signal of 70 K should be higher than that of 44K. Thus, my estimate looking at the S2 images do not match with the intensity plot shown in Fig. 2d, under the assumption that the sample did not shift with cooling. Alternatively, is the sample shifting as it is being cooled? Fig. 2a (S1 images) show outlines drawn for the flake at all temperatures, and in this case the sample seemed to have moved.

To minimize confusion, I recommend the following.

- Explicitly explain the origin of the intensity changes at 44 K for the S2 image, whether this is due to sample shift or not.
- Draw the 3 dashed boxes (1L, 2L, and 3L) from which the S2 average intensity was calculated for all temperatures, so that it is easy to identify these regions for all temperatures.
- Can the authors actually calculate the S2 average intensity from larger boxes? The rectangular boxes are much smaller than the areas of 1L, 2L, and 3L.

The assignment of weak metallicity for 2L and weakly insulating behavior for 1L rests on Fig. 2b (normalized S2 images) and Fig. 2d (averaged S2 intensity from the rectangular boxes). Therefore, this seeming

discrepancy as I explain above needs to be clearly explained.

#2. Fig. 2e mentioned in the text should be referred as Fig. 2d.

#3. Image blur due to SPP

Blurring of the image at 44 K (Fig. 2a, b) is explained by the SPPs that have a spatial extent of 8 – 16 microns in the 3L, metallic region. This seems reasonable. But I also think that the 2L region is blur at 44 K. (note: I am thinking the 2nd brightest region at 44 K corresponds to 2L, not 1L, assuming the sample moved. Or, is the 2nd brightest region at 44 K actually the 1L region? This comment is related to comment #1.).

Thus, can the authors explain why the 2L region also gets blurred? Can SPPs be supported there as well due to the weak metallicity and the estimated carrier density? Or is there blurring? The image appears so, but some line profiles that show the broadening (or the lack of broadening) of features would be convincing, and why it happens for the 2L region.

#4. Figure 4c.

Authors plot the experimental values of the S_2 of bilayer normalized to S_2 of substrate as a function of temperature (orange squares) with the simulated data at various gap sizes. This plot convincingly shows that the bi-layer case should have a gap that is between -20 meV and -10 meV, which also supports the weak metallicity of the bilayer case.

Can the authors also plot the experimental values for the monolayer region in the right panel of Fig. 4c?

Reviewer #3 (Remarks to the Author):

The authors apply the AFM-focused THz microscopy technique that goes beyond diffraction limit to study the THz response of the electronic properties in mono- and few- layered WTe₂. It is concluded a metal behavior of tri-layer and semiconductor behavior of mono layer. The results are compelling and the article is well-written. To make the work publishable at Nat Commun here are the following points the authors shall address:

a) The technique is not conventional. To make the results even more convincing, the same technique shall be applied to "bulk" samples (> 3 layers), and discuss the difference between the trilayer metallic states and bulk metallic states. The bulk states will serve as a reference state, from which all other conclusions can be drawn with greater confidence. In particular, does Weyl states emerge in trilayer sample?

b) The nature of bilayer WTe₂, which is probably the most intriguing part, is still unclear, at least based on the current set of theoretical inference. Why only fix T=40K? The authors are strongly urged to improve the theoretical scattering model or goes to lower temperatures, with possible more data to draw a more concrete conclusion on the electronic nature on bilayer.

REVIEWER COMMENTS

Reviewer #1 (Remarks to the Author):

1. The authors apply a (fairly) novel technique to exfoliated WTe₂ flakes in the 1-3 monolayer regime. Much of the work addresses validation of the method in the context of this material system.

The data the authors receive are impressive, especially given the novelty of the technique. The data requires, however, modeling to gauge the importance of lateral surface plasmon polaron scattering, which render the 'steps' the authors describe far more gradual than maybe desired. The problem here is NOT one in which the data obtained by the authors is questionable, but rather one, in which edge effects – and indeed many localized effects this technique would otherwise be a great tool to study – are obscured.

>> We thank the reviewer for the comment. Near-field modeling is indeed an important piece of proof of surface plasmon polariton (SPP) in this paper. The necessity for the modeling is that the THz range SPP in few-layer WTe₂ has ultra-long wavelength which is comparable to or longer than the typical scale of the sample. In particular, the wavelength of the SPP is 6 – 20 μm on 3L WTe₂ and the typical length scale of tri-layer region is 5~10 μm . The consequence is that the fringe patterns of propagating or standing wave SPP observed in other samples are absent [Ni, et al. Nature 557.7706 (2018): 530-533.]. Instead, the pattern of SPP is dependent on the shape of the sample. To investigate the effect of SPP, we used two methods, reflectivity $r_p(\omega, q)$ calculation (Fig. 3a and 3b) and real-space SPP simulation (Fig. 6), to model the near-field signal and real-space pattern, respectively. Both modeling results support the ultra-long wavelength SPP existing in the tri-layer WTe₂.

Figure R1| Nano-THz image of 1L, 2L and 3L WTe₂. **a** Near-field S_1 image of the whole sample. The near-field response of 1L is close to the insulating substrate because of its >60 meV band gap. The blurred edge is not observed on 1L. **b** Zoomed-in near-field image at 250 K. A 200nm wide artifact is indicated by black arrows. The resolution of the near-field tip is fine enough to clearly resolve this line-shaped artifact.

We remark that the gradual steps, as identified by the reviewer, is indeed one of the edge features present in few-layer WTe₂ systems. The gradual steps are not present in the monolayer region

because monolayer WTe_2 is insulating (Fig. R1). The near-field technique is indeed ideal to investigate many localized features in THz range. However, the conductance of the topological insulator edge mode in mono-layer WTe_2 is too low to detect a discernable contrast in near-field. The clear identification of the edge state in THz range would need gateable devices in future nano-THz experiments.

2. What is left is a characterization of the material across an interesting thickness range that is complementary to existing transport and spectroscopic measurements. Enhanced utility of their technique derives maybe from applicability to smaller samples. There is little or no concern that the data obtained by the authors is valid.

>>We appreciate the reviewer's comment about validity of the data of our experiment. We wish to emphasize that there is currently no conclusion of the gapped or gapless nature of bilayer WTe_2 . The inherent small energy scale (~ 10 meV) associated with this problem in combination with the small physical size (<10 μm) of the bilayer WTe_2 samples pose extreme challenges on the experimental verifications. To the best of our knowledge, our low-temperature nano-THz imaging experiment provides the first real space evidence of the metallicity of bilayer WTe_2 , strongly suggesting a negative gap picture that is in contrast with transport studies. Since nano-THz is a local probe of metallicity and no electrical contacts are needed on the sample, we believe our results reveal more intrinsic response of bilayer WTe_2 .

3. Overall, the manuscript offers information on the films that is well in line with common expectations. There is very little new or outstanding about the paper, other than the application/development/validation of a new technique. This paper should definitely be published. It ought to be an editorial decision, whether Nature Communication is the proper venue for data as described above.

>> We appreciate the reviewer for recognition of the novel technical development in our manuscript. As mentioned above, we believe our technique has the unique advantage of probing local and intrinsic electromagnetic behavior on the sample surface. Therefore, the result provides strong evidence of intrinsic metallicity of bi-layer WTe_2 and lays the foundation for probing band alignment of similar materials with small gap sizes. In response to and Ref. 3's comment (a), we also performed new experiments on thicker WTe_2 samples and confirmed that the gap size of 3L – 12L WTe_2 can be reasonably considered as $\Delta \sim 20$ meV. The content corresponds to the Fig. 5 in the revised manuscript.

Reviewer #2 (Remarks to the Author):

Authors report THz range EM behaviors of WTe_2 mono-, bi-, and tri-layers, encapsulated in

hBN, as a function of temperature. Authors observe / conclude a metallic state and SPPs for the tri-layer region, moderate metallicity for the bi-layer region, and weakly insulating behavior for the monolayer region. The main data is presented in Figure 2 and the manuscript relies heavily on modeling to understand the data. I have several technical comments that I would like the authors to address.

#1. Seeming discrepancy between Fig. 2d and the S2 image (Fig. 2b).

The S2 image at 44 K looks shifted vertically. It looks shifted because the feature corresponding to the 2L region becomes suddenly dark at 44 K and the 1L region becomes brighter at 44 K. Thus, comparing the 1L region of 150 K versus 44 K for example, the normalized S2 averaged signal of 44 K should be higher than that of 150 K based on the images. And for 2L region, the normalized S2 averaged signal of 70 K should be higher than that of 44K. Thus, my estimate looking at the S2 images do not match with the intensity plot shown in Fig. 2d, under the assumption that the sample did not shift with cooling. Alternatively, is the sample shifting as it is being cooled? Fig. 2a (S1 images) show outlines drawn for the flake at all temperatures, and in this case the sample seemed to have moved.

>>We thank the reviewer for the careful review of our manuscript. The S2 image at 44K is shifted vertically because the mono-layer are zoomed in and imaged independently at 44K. We now add label to the 44K image to clarify the layer numbers to avoid confusion. Previously the image of the monolayer was included in Supplementary Note 3 (Fig. S3), we now add to Fig. 2c in the revised main text.

Here we show the updated figure in Fig. R2 for reference. Fig. 2 in the main text is also updated in the revised manuscript.

Figure R2| Temperature dependent near-field maps of nano-THz response of WTe₂ micro-crystals. The scale bars in all panels are 3 μm. **a** Near-field images of the normalized amplitude contrast S_1/S_1^{sub} of nearly

identical regions at 6 different temperatures between 250 K and 70 K. The 44K image include only 2L and 3L regions and the 1L region is shown in Fig. 4c. Broadband THz signal utilized in these images shows intensity peaked at 0.6 THz. S_1^{sub} : the signal due to the SiO₂/Si substrate has negligible temperature dependence in the THz range studied here. The horizontal dashed arrow in the panel at 200 K indicates the scanning line-cut used to construct the plot in Fig.2d. Micrometer-sized dark spots also visible in the topographic AFM contrast can be attributed to bubbles in the encapsulated structures. **b** Near-field S_2/S_2^{sub} images taken simultaneously with S_1/S_1^{sub} . **c** Enlarged images zoomed at the interface between 1L and 2L, 1L and Substrate. **d** S_1/S_1^{sub} line-cut (averaged over 5 neighboring pixels) at 250 K and 70 K. The line-cut corresponds to the arrow in Fig.2a. **e** Normalized S_2 signals averaged in the regions indicated in the S_2 images (white dashed boxes) for the substrate and for 1L, 2L, 3L regions of WTe₂. The filled squares are experimental data. The dashed line is model calculation based on the dielectric property of the bulk material. The absolute value is rescaled to be comparable with 3L WTe₂ based on bulk WTe₂ dielectric function.

To minimize confusion, I recommend the following.

- Explicitly explain the origin of the intensity changes at 44 K for the S2 image, whether this is due to sample shift or not.
- Draw the 3 dashed boxes (1L, 2L, and 3L) from which the S2 average intensity was calculated for all temperatures, so that it is easy to identify these regions for all temperatures.
- Can the authors actually calculate the S2 average intensity from larger boxes? The rectangular boxes are much smaller than the areas of 1L, 2L, and 3L.

We thank the reviewer for the suggestions. We made corrections accordingly.

>>The 44K images in Fig. 2a and 2b are shifted vertically. Labels of layer number are added to avoid confusion. This is not due to sample drift. The image of mono-layer region at 44K is shown in Fig. 2c in the main text. The updated Fig. 2 is displayed above and included in the revised manuscript.

>> We added dashed boxes to all images in Fig. 2b to indicate the region used for average. Due to the lack of monolayer at 44K in Fig. 2a and 2b, we use Fig. 2c to calculate the near-field signal on mono-layer. The data box used for analyzing is also indicated in Fig. 2c.

>> In Fig. S4 of the revised Supplementary Note 4, the signal from each region is averaged within the whole region area. Previously the purpose of averaging in smaller areas is to avoid including bubbles and gradual edge effect. The areas of regions are indicated in Fig. 2a. Here, we also show this image (Fig. R3) for reference.

Figure R3 | The data points are normalized near-field signal S_2/S_2^{sub} averaged within the entire region of the substrate, 1L, 2L and 3L in Fig. 2a, 2b and 2c in the revised main text.

We can use larger boxes to average the near-field data. In Fig. R3, we show the normalized near-field signal-averaged within the entire regions identified in Fig. R2a, R2b and R2c. The change of the averaging method does not change the conclusion.

The assignment of weak metallicity for 2L and weakly insulating behavior for 1L rests on Fig. 2b (normalized S2 images) and Fig. 2d (averaged S2 intensity from the rectangular boxes). Therefore, this seeming discrepancy as I explain above needs to be clearly explained.

>> We thank the reviewer for the question. The seeming discrepancy is due to the fact the zoomed-in image of 1L was previously displayed in the Supplementary Note 3 Fig. S3 only. Now we add them to the main text (Fig. 2c). Combining 44K images in Fig. 2b and 2c, we show how the data in Fig. 2d is collected. Therefore, the seeming discrepancy is addressed in the revised manuscript and we thank the reviewer again for careful reading of our manuscript.

#2. Fig. 2e mentioned in the text should be referred as Fig. 2d.

>> We thank the reviewer for the correction. The mistake has been corrected in the revised manuscript.

#3. Image blur due to SPP

Blurring of the image at 44 K (Fig. 2a, b) is explained by the SPPs that have a spatial extent of 8 – 16 microns in the 3L, metallic region. This seems reasonable. But I also think that the 2L region is blur at 44 K. (note: I am thinking the 2nd brightest region at 44 K corresponds to 2L, not 1L, assuming the sample moved. Or, is the 2nd brightest region at 44 K actually the 1L region? This comment is related to comment #1.).

>> We thank the reviewer for the careful reading of our manuscript. The second brightest region in 44K image is 2L region. The field of view in 44K image is shifted vertically compared to other temperatures. The 1L images are added to Fig. 2c.

Thus, can the authors explain why the 2L region also gets blurred? Can SPPs be supported there as well due to the weak metallicity and the estimated carrier density? Or is there blurring? The image appears so, but some line profiles that show the broadening (or the lack of broadening) of features would be convincing, and why it happens for the 2L region.

>>We appreciate the question from the reviewer. The short answer is that the 2L region is indeed impacted by ultra-long wavelength surface plasmon polariton (SPP). We reached this conclusion by analyzing Fig. 6 in the main text (display here as Fig. R4).

Figure R4 | **a, b** Modeling of real-space near-field (S_1/S_1^{sub}) pattern associated with SPP at 295 K (**a**) and 70 K (**b**) on the investigated sample geometry. Dashed lines highlight the physical boundaries of all regions. Line-cuts across bi- and tri-layer WTe₂ are indicated by the solid line in both images. **c** Model S_1/S_1^{sub} line-cuts extracted from **a** and **b** are plotted in solid lines. Along with the model result, experiment line-cuts are displayed in dashed lines and are shifted vertically for clarity.

We firstly use near-field model to conclude that 2L WTe₂ is a semi-metal with ~ 10 meV overlap between the conduction and valence band. Then we carried out real-space modeling of SPP pattern on all regions in the end of the paper (Fig. 6). By comparing the rea-space modeling results with the experiment images, we conclude that the gradual change of signal in 2L WTe₂ is also caused by SPP. The wavelength of 2L WTe₂ is 4.2 μm .

Now In order to avoid confusion, we stress the existence of this blurred signal on 2L in the first half of the paper (the 3rd paragraph of page 6 of the revised main text).

#4. Figure 4c.

Authors plot the experimental values of the S2 of bilayer normalized to S2 of substrate as a function of temperature (orange squares) with the simulated data at various gap sizes. This plot convincingly shows that the bi-layer case should have a gap that is between -20 meV and -10 meV, which also supports the weak metallicity of the bilayer case.

Can the authors also plot the experimental values for the monolayer region in the right panel of Fig. 4c?

>>We thank the reviewer for the suggestion. Fig. 4c in the main text is indeed more informative with 1L WTe₂ data added, see Fig. R5 below. In addition, we also made modeling for 3L WTe₂. Now, 3L model results and the experiment data are also displayed for comparison. We have updated the

figure in the revised manuscript and the new Fig. 4 is also reproduced below for reviewers' convenience.

Figure R5| Near-field electro-dynamics of thermally activated carrier of few-layer WTe₂. **a** Temperature dependence of thermally activated carrier densities at different gap sizes calculated (Supplementary Note 5) based on the band structure investigated by ARPES [12]. **b** Near-field spectroscopic response of the thermally activated carriers of a model 2L WTe₂ with $\Delta = -10 \text{ meV}$. The green shaded region represents the power spectrum of the THz probe. **c** Right panel: Temperature dependent WL signal calculation based on LRM for 1L (black curve), 2L (blue curve) and 3L (red curve) WTe₂ with gap sizes 60 meV, -10 meV and -20 meV respectively. Along with the model, nano-THz data of 1L, 2L and 3L WTe₂ are displayed with squares. Both the model curves and experiment points are normalized to the substrate value. Left panel: The gap-size dependent near-field signal of 2L (blue) and 3L (red) at 44 K. The signal level is strongly suppressed when the gap is close to zero or positive. **d, e** Hypothetical band structure of semimetallic 2L WTe₂ (left) and insulating 1L WTe₂ [12] with a bandgap $\Delta > 60 \text{ meV}$ (right).

Reviewer #3 (Remarks to the Author):

The authors apply the AFM-focused THz microscopy technique that goes beyond diffraction limit to study the THz response of the electronic properties in mono- and few- layered WTe₂. It is concluded a metal behavior of tri-layer and semiconductor behavior of mono layer. The results are compelling and the article is well-written. To make the work publishable at Nat Commun here are the following points the authors shall address:

a) The technique is not conventional. To make the results even more convincing, the same technique shall be applied to "bulk" samples (> 3 layers), and discuss the difference between

the trilayer metallic states and bulk metallic states. The bulk states will serve as a reference state, from which all other conclusions can be drawn with greater confidence. In particular, does Weyl states emerge in trilayer sample?

>>We appreciate the reviewer for recognizing the importance of our work and for the suggestion of measuring thicker WTe_2 samples. We now extend our measurement on a different sample with regions of different thicknesses (3L, 4L, 6L, 12L). The result is shown in Fig. R6 (Fig. 5 in the revised main text).

The data acquired on the new sample is displayed as square points and the data extracted from the old sample is in cross points. 3L WTe_2 regions exist on both samples so we can compare the two datasets directly. The agreement of the signal level on 3L in both experiments indicates the good consistency of our nano-THz measurements. We then tried to use the lightning-rod model to calculate the near-field signal on 3L – 12L WTe_2 with only one free parameter, the gap size Δ . The model curves are displayed as dashed lines. When $\Delta \sim -20$ meV. the model simultaneously matches the high temperature (red) and low temperature (blue) data. We conclude that from 2L to 3L, the negative gap continues to grow in the negative direction. In addition, from 3L – 12L, the signal can be reasonably described by a semimetal with a negative gap. We remark that at 12L the THz near-field signal is already 86% of the bulk limit (44K). The fact that 3L – 12L can be captured with a single model strongly supports the claim that the property of 3L WTe_2 is already close to the bulk material in terms of THz electrodynamic properties. Moreover, the clear deviation of the signal level for the 2L and 1L results from the bulk model demonstrate their less metallic and insulating behaviors, respectively.

Figure R6| Layer-dependent white-light near-field signal on WTe_2 . **a** Topography image of the sample. The boundaries of different regions are indicated with dashed lines. **b** The S_2/S_2^{sub} image taken simultaneously with **a**. **c** Layer dependent near-field signal on WTe_2 . The experiment data are collected on two different samples. The data points extracted from Fig. 2 are marked with cross. The data points acquired on the new sample is marked with square. 3L WTe_2 is measured on both samples. High and low temperature data are displayed in red and blue points. The dashed lines are LRM near-field signal contributed by thermally activated carrier with gap size ~ -20 meV at 44 K (blue) and 295 K (red).

>> We thank the reviewer's question about whether Weyl states emerges in tri-layer sample. As for the Weyl semimetal nature in 3L WTe₂, at the moment we cannot make any conclusions since the Weyl points in WTe₂ are located well above the Fermi level (~50 meV, Soluyanov, et al. Nature 527.7579 (2015): 495-498.) Since the THz probe we use are limited to 1.5 THz or 6 meV, the charge dynamics is hardly affected by the existence of Weyl points.

b) The nature of bilayer WTe₂, which is probably the most intriguing part, is still unclear, at least based on the current set of theoretical inference. Why only fix T=40K? The authors are strongly urged to improve the theoretical scattering model or goes to lower temperatures, with possible more data to draw a more concrete conclusion on the electronic nature on bilayer.

We appreciate the reviewer for recognizing the significance of the intriguing nature of bilayer WTe₂. The 40 K temperature limit is mainly an experimental limitation associated with the specific design of our nano-THz microscope. Due to the requirement of efficient THz focusing and signal collection, we take the advantage of the large focusing solid angle. Therefore, the heat-shield cannot be conveniently applied and the base temperature of the experiment was limited to 40K. We are working on reaching lower temperatures for our state-of-the-art low temperature nano-THz microscope.

Finally, we reiterate our points that we have strong evidence that bilayer is weakly metallic, this is contradictory to transport studies, but we believe our local nano-THz probe reveals a more intrinsic electrodynamic response of bilayer WTe₂, since we are not affected by extrinsic effects such as defects and contacts which may be an issue in transport. This is of course not a complete understanding, but significant enough information gained through a novel experimental approach.

REVIEWERS' COMMENTS

Reviewer #2 (Remarks to the Author):

The authors have adequately addressed my questions and suggestions.

Reviewer #3 (Remarks to the Author):

The authors performed extensive extra measurements including thicker samples, the data looks consistent showing the approaching to bulk behavior. The metallic bi-layer part which reveals intrinsic EM response is a highlight, and the work deserves to be published in Nat Commun.

REVIEWERS' COMMENTS

Reviewer #2 (Remarks to the Author):

The authors have adequately addressed my questions and suggestions.

→We thank the reviewer for the careful review and acceptance of the response.

Reviewer #3 (Remarks to the Author):

The authors performed extensive extra measurements including thicker samples, the data looks consistent showing the approaching to bulk behavior. The metallic bi-layer part which reveals intrinsic EM response is a highlight, and the work deserves to be published in Nat Commun.

→We thank the reviewer's recognition of our effort to acquire new data and recommendation of the publication.